# Environmental Credit Constraints and the Enterprise Choice of Environmental Protection Behavior

**Chunrong Yan [1], Xintian Xiang [1]** **, Liping Li [2,\*] and Guoxiang Li [3]**

[1] School of Economics and Management, Shanghai Polytechnic University, Shanghai 201209, China; cryan@sspu.edu.cn (C.Y.); xxt265@163.com (X.X.)

[2] School of Economics and Social Welfare, Zhejiang Shuren University, Hangzhou 310015, China

[3] School of Business, Nanjing Normal University, Nanjing 210046, China; lgx_0614@126.com

\* Correspondence: llp987789sufe@163.com

**Abstract:** Choosing appropriate environmental protection strategies is important in improving enterprises' economic and environmental performance. Based on the data of A-share listed enterprises from 2009 to 2019 in China, this paper uses the difference-in-differences model to identify the effects of environmental credit constraints on the enterprise choice of environmental protection behavior. We find that environmental credit constraints motivate some enterprises to choose active environmental behavior due to the incentive effect of environmental credit constraints on R&D investments. However, some enterprises may adopt evasive strategies because environmental credit constraints increase production costs and debt. State-owned enterprises prefer active environmental protection strategies to address environmental credit constraints, while private enterprises mainly adopt evasive strategies. Environmental credit constraints make high-interest and high-profitability enterprises choose active environmental strategies. Environmental credit constraints generated by enterprises' evasive environmental behavior increase the probability of litigation and arbitration cases, and environmental credit system construction in the short term may exacerbate unemployment, which the government needs to pay attention to when developing and implementing a blacklist system for environmental fraud. Although there are limitations in this paper in terms of research objectives and samples, the results are important for improving the environmental management system and the operating performance of enterprises.

**Keywords:** environmental credit; R&D investment; environmental protection

## 1. Introduction

The construction of an ecological civilization has put forward higher requirements for environmental quality. Given rapid economic development and population growth, environmental problems are becoming increasingly prominent [1,2], affecting the sustainable development of the economy and society. To promote high-quality economic development, the Chinese government has gradually improved laws and regulations in the field of environmental protection to promote environmental protection and an ecological civilization, strengthen environmental constraints on highly polluting enterprises, and increase penalties for environmental fraud [3]. In addition, China is actively promoting green and low-carbon development, comprehensively promoting the carbon peaking and carbon neutral strategy, encouraging green technology innovation, increasing the proportion of clean and renewable energy, and reducing energy consumption and pollution emissions [4]. The environmental credit system is an evaluation and filing mechanism based on enterprises' environmental protection behavior that aims to promote the fulfillment of environmental obligations and strengthen environmental management through the evaluation and public disclosure of such enterprise behavior [5]. The construction of an environmental credit system to promote environmental governance can influence the performance of enterprise

environmental governance by improving several aspects, such as the awareness of enterprise environmental responsibility, environmental investments, environmental image, and competitiveness.

Faced with environmental credit constraints, different types of enterprises may adopt differentiated environmental protection strategies. On the one hand, a strong sense of social responsibility makes enterprises adopt survival and long-term development as their ultimate goal and consciously assume various social responsibilities, including pollution prevention and control. Driven by a business motivation, enterprises with investment strength and applicable technology may choose to improve the environmental attributes of their products and reduce pollution emissions and resource consumption through technology innovation, green design, and clean production [6], thereby meeting environmental requirements and fulfilling environmental responsibilities [7]. Enterprises can actively launch environmentally friendly brands or eco-products, such as public welfare activities and environmental education, to enhance their image and market value. On the other hand, enterprises may choose to ignore the pressure from environmental constraints and not take active measures to address environmental issues because they must compensate for internalized environmental costs using the revenues of their main business [8]. Therefore, some enterprises often lack the willingness and ability to engage in environmental governance. Especially regarding green technology innovation activities, such as end-of-pipe and source management, the high risk and cost of innovation inhibit enterprises' motivation to engage in green innovation [9], forcing them to adopt negative environmental strategies and respond to environmental policies by laying off employees and reducing production. What environmental strategies will the construction of an environmental credit system cause enterprises to choose?

In this paper, the numbers of green patents and separations are selected to measure active and evasive environmental behaviors, respectively, and the impact of environmental credit constraints on enterprise environmental behaviors is studied using data on China's A-share listed enterprises from 2009 to 2019. In this paper, we mainly answer the following questions: How do environmental credit constraints affect enterprises' environmental protection behavior? What are the working channels? Are the effects of environmental credit constraints influenced by other factors? The marginal contributions of this paper are as follows: (1) Although existing studies have pointed out that environmental policies can have some influence on enterprises' environmental protection behavior [10], they do not provide an explanation from the perspective of environmental credit constraints. Based on the intrinsic relationship between environmental credit system construction and enterprise pollution management, we study the choice of an enterprise environmental protection strategy from the perspective of environmental credit constraints, analyze the environmental protection strategies that different types of enterprises should choose when facing environmental credit constraints, and clarify how different environmental protection behaviors are formed. (2) In this paper, we fully discuss the effects of property rights attributes, external interest associations, and differences in profitability in the policy implementation process to further clarify the differences in the motivation for environmental behavior given the heterogeneity of enterprises to provide a reference for enterprises to better optimize their environmental protection strategies and explore more precise and effective incentive programs. The results of this study can help guide enterprises to transform evasive environmental protection into active environmental protection and effectively carry out green innovation activities.

## 2. Literature Review

Enterprises' responses to environmental pollution problems largely depend on their organizational environment, both external and internal [11]. The external environment includes laws, policies, community public, industrial markets and capital markets, etc. Many scholars have conducted in-depth research on environmental regulations and demand guidance [12]. First, environmental regulations. Enterprises' environmental protection be-

havior mainly meets the requirements of the government's environmental regulations and sustainable economic development. The government formulates relevant environmental protection systems, laws, and regulations to restrain enterprises' production behavior, promote their energy conservation and emissions reductions, and reduce pollution emissions and investments to increase environmental protection [13]. Environmental regulations have a significant positive impact on enterprise technological innovation [14]. Reasonable government regulations can encourage enterprises to increase R&D investments and reduce pollution through green technological progress to lower enterprises' production costs and achieve green development [15,16]. However, in the face of environmental constraints, some enterprises adopt the pollution transfer method to evade or reduce punishments, which increases pollution in the industry's import areas. At this time, environmental regulations may have an innovative compensation effect to reduce the probability of pollution transfers [17]. Some scholars also believe that environmental regulations have a negative impact on enterprises' environmental behavior. Environmental regulations increase the cost of environmental governance, reduce productivity and profitability, and lead enterprises to engage in more negative environmental protection behaviors [18]. Second, demand guidance. Consumer demand has a very important position and plays a very important role in market demands [19]. Economic and social development and improvements in residents' incomes, consumers' awareness of environmental protection consumption, and the expectation of environmentally friendly products have an impact on enterprises' environmental protection behavior [20]. Because enterprises carry out production and business activities based on consumer demand, environmentally friendly product preferences can prompt them to seek green upgrade paths [21,22]; enterprises, to gain greater market share, also strengthen green product production through environmental protection investments [23] to meet market demand through an appropriate environmental protection strategy [24].

The internal environment involves internal policies, organizational structures, corporate values, employee attitudes and beliefs, organization resources and capabilities, etc. Given the change in the weight of the factors affecting enterprises' environmental behavior, their internal factors become increasingly important to their environmental behavior, in which enterprise scale and management's environmental awareness are the two most important factors [25,26]. Compared with small- and medium-sized enterprises, large enterprises emit more pollution and are more vulnerable to environmental regulations. Therefore, large enterprises take more active environmental protection actions on the premise of responding to policies [27]. The human factor that complies with the pollution problem caused by enterprises' production and operation activities, in addition to enterprise factors, cannot be excluded [28], and management's support of enterprise environmental behavior is of great importance [29]. If enterprise management has a strong sense of environmental protection and understands and accepts the importance of an environmental management strategy for the entire organization, it will increase inputs for environmental pollution controls, optimize the production process and green technology innovation, and improve the efficiency of energy resource utilization and the pollutant treatment rate, thus producing good environmental performance [30].

Some scholars have studied the impact of the credit system on environmental governance [31]. The essence of the market economy is the credit economy, and the credit system is developed along with studying environmental pollution problems [32,33]. From the perspective of environmental governance, the establishment of and improvements in credit systems can provide a theoretical basis for environmental governance behavior [34]. Enterprises are the main body of environmental governance. A good credit system can expand enterprises' external financing channels, effectively promote green technology innovation, increase the output of green products, and reduce pollution emissions in enterprises' production processes to achieve better environmental governance effects [35]. Given the environmental problems caused by rapid economic development, the government has paid increasing attention to the construction of an environmental credit system. The environmental credit system will place strong environmental constraints on pollution-intensive



enterprises [36], forcing them to improve their environmental reputations to obtain continuous policy support from the government, reduce environmental costs through energy conservation and emissions reductions, and reduce pollutant emissions at the source [37]. The environmental credit system is also helpful in improving enterprises' environmental information disclosures, environmental awareness, and credit and environmental management levels and urges them to consciously fulfill their environmental responsibilities to obtain good environmental credit and enhance the convenience of obtaining financing, mortgage guarantees, project approvals, and other aspects [38].

Enterprise environmental governance is a key topic in academia. In summary, changes in enterprises' organizational environments can affect their choice of environmental behavior. As a soft environmental constraint, the credit system also has an impact on environmental governance by forcing enterprises to reduce their pollution emissions, reduce environmental costs, and improve their environmental reputations, as well as obtaining more government support. However, existing studies have rarely studied enterprises' environmental behavior from the perspective of the construction of an environmental credit system and the differences in the strategies selected by different types of enterprises related to their environmental behavior. Moreover, existing studies have not explored how environmental credit constraints influence enterprises' environmental behavior, leaving room for further research in this paper.

## 3. Theoretical Hypothesis

Economists, represented by Porter, have argued that environmental policies can increase the productivity; they have developed the important hypothesis of Porter that appropriate environmental regulation can increase productivity by incentivizing innovation, efficiency, and internal redistribution. As the famous Porter hypothesis postulates, environmental regulations can directly or indirectly reduce pollution emission [39]. Given the constraints of environmental credit, the government sector uses polluters' credit records as an important basis for determining the frequency at which they supervise environmental protection [40], implement refined enterprise management, strengthen the environmental supervision of highly polluting enterprises, and incorporate enterprises' environmental pollution behavior into the national credit management information system to allocate economic resources and support policies accordingly. Environmental credit constraints affect enterprises' environmental governance input, product R&D, and environmental strategic decisions, etc. [41], force enterprises to choose appropriate environmental protection strategies and avoid environmental penalties while ensuring economic output, and guide enterprises to carry out environmental protection and pay attention to environmental performance. Enterprise environmental protection behavior may be of the following three types [42]. First, active environmental protection behavior (AEB). Some enterprises believe that the construction of an environmental credit system can bring new opportunities for their development [43]. Given environmental credit constraints, such enterprises will pay more attention to consumer preferences, actively research and develop environmentally friendly products, gradually increase the share of sales of such products, and improve their competitiveness through product differentiation strategies. Enterprises will also increase their investments in environmental pollution control, especially to prevent and control sources of pollution, and rely on environmental technological progress to solve the pollution problem [44]. Second, passive environmental protection behavior. Enterprises adopting this behavior will have to pay pollution control fees to improve their production modes because of environmental credit constraints, which is bound to further increase production costs. Such enterprises may choose to reduce their production scale but continue to use old manufacturing production processes, reducing pollutant emissions by reducing economic output [45]. Third, evasive environmental protection behavior (EAD). Enterprises adopting this behavior have a negative attitude toward environmental credit constraints and will take advantage of the differences in the environmental regulation

intensity in different regions to avoid environmental penalties or reduce environmental costs through production transfers [46].

**Hypothesis 1.** *Environmental credit constraints can produce active environmental protection behavior for enterprises while reinforcing evasive environmental protection behavior.*

What is the channel through which environmental credit constraints affect enterprises' environmental protection behavior? First, R&D investments. One environmental protection action strategy of enterprises is to promote the transformation and upgrading of production equipment and technology by increasing investments in R&D to improve the innovation ability of green technology. There is a significant positive correlation between the output and input of enterprise green innovation. To realize green production, enterprises will increase emission reduction inputs throughout the production process and promote biased technological progress related to energy savings and emission reductions, thus reducing long-term environmental costs. R&D investments are necessary for enterprises' green technology innovation [47]. An increase in R&D investments can enrich the resources available for enterprise R&D, creating favorable conditions for green technology innovation. In addition, some of enterprise's R&D investments will flow to universities and relevant R&D departments through R&D cooperation, generating a greater knowledge spillover effect. As an important environmental regulation method, the environmental credit system will have an important impact on enterprises' environmental protection behavior. On the one hand, the construction of an environmental credit system can expand the financing channels of enterprises with good credit, provide more convenient services to enterprises through flexible credit supervision, enhance the driving force of green technology innovation, increase R&D investments in environmental protection, and encourage enterprises to take more active environmental protection actions. On the other hand, the environmental credit system is also a kind of soft constraint, forcing enterprises to strengthen their ability to control pollution through technological innovation and reduce environmental fraud to obtain more government policy support and enterprise credit by improving environmental credit. Second, debt. Regarding the financing environment, enterprises with better financing ability will have diversified financing channels and more abundant funds to support technological innovation activities. Regarding credit audits, when making decisions financial institutions represented by banks must focus on the degree to which enterprises focus on a green environment and other indicators. According to the Green Credit Guidelines requirements, commercial banks must strictly control financing approval thresholds, use enterprises' environmental performance as an important reference factor for financing approvals, and limit high-polluting enterprises' access to capital or issue high-interest loans to them to increase their financing pressure and expand their debt [48]. This will force financial institutions to pay attention to enterprises' environmental behavior when approving financing applications and to use the environmental risks of an enterprise's business as the main reference point. In this way, the construction of an environmental credit system will force enterprises to disclose environmental information. If enterprises' environmental risks increase, the risks borne by financial institutions and creditors will also increase, thus increasing the financing costs and debt of such enterprises [49]. Enterprises that engage in EAD may adopt strategies such as reducing their production and eliminating employees to reduce financing and production costs, avoid environmental fraud, and affect their enterprise credit as much as possible.

**Hypothesis 2.** *For enterprises that adopt active environmental protection behavior, environmental credit constraints will encourage them to increase their R&D investments to promote green technology innovation.*

**Hypothesis 3.** *Environmental credit constraints will increase corporate debt and force some enterprises to adopt evasive environmental protection behavior.*

## 4. Research Design

### 4.1. Variable Selection

(1) Explained variable. The explained variable in this paper is enterprise environmental behavior. In a broad sense, enterprise environmental protection behavior is not only limited to the product manufacturing process but also includes a series of enterprise environmental protection behaviors that can help enterprises obtain environmental rationality and maintain stakeholder relationships and are usually divided into two categories: substantive and symbolic environmental protection behaviors. The former refers to the environmental protection behavior related to the core business of the enterprise and can significantly improve its environmental performance. The latter refers to behavior that is not related to the main business but can help companies establish or maintain an environmental protection image. The enterprise environmental behavior studied in this paper is mainly substantive environmental behavior and includes the following two types. First, active behavior (AEB). AEB refers to reducing the environmental damage of enterprise production by improving its pollution control capacity. The core of this behavior is reflected in improvement in enterprises' ability to engage in green technology innovation. The number of green patent applications is adopted as a measure of green technology innovation. Patent types mainly include appearance design patents, utility model patents, and invention patents. This paper focuses on research on the ability of enterprises to engage in emission reduction technology innovation, including terminal governance technology, energy saving and emission reduction technology, and clean production technology, etc. Therefore, the sum of the number of green invention patents and green utility model patent applications (the unit is PCS) was selected to measure enterprises' green technology innovation. Second, evasive behavior (EAD). EAD refers to enterprises reducing the number of employees within a reasonable range to reduce the emission of pollutants by reducing economic output. The number of dropouts and resignations of enterprises (the unit is people) was adopted to describe EAD. This is because for EAD enterprises, one of the effective ways to reduce pollution emissions in the short term is to reduce the production scale, which is bound to reduce the demand for workers and forces enterprises to eliminate some employees with low labor skills or low work efficiency. The data come from the State Intellectual Property Office, the annual report of listed companies, and the Wind Information Database, etc.

(2) Explanatory variables. We use the existence of environmental credit constraints as the independent variable. We approximate the presence of environmental credit constraints by relying on the blacklist system that policymakers use to promote environmental protection initiatives. Environmental watchdogs depend on this framework to record and publicize violations committed by enterprises, institutions, and individuals. Polluting enterprises are the focus of the blacklist system, and governmental bodies and social organizations share information and carry out joint punishment. The scheme covers a relatively broad range of activities but attributes particular attention to enterprises' compliance with general environmental regulations, including laws, rules, normative documents, and environmental standards, as well as the extent to which polluting enterprises fulfill their environmental social responsibilities in production and operation activities. We rely on official city websites to manually collect environmental regulation-related data and record the decisions issued by environmental watchdogs, such as the National Development and Reform Commission and other governmental agencies. We subsequently assume that environmental credit constraints have been imposed on local enterprises if cities have developed and implemented the measures mandated in the context of the blacklist framework. The data are obtained from the websites of the people's governments of various cities in China and from government websites such as the Development and Reform Commission and the Department of Ecology and Environment.

(3) Control variables. Referring to the variable selection criteria of [50,51], enterprise- and city-level indicators (includes 2659 enterprises, 226 cities) are selected as the control variables. The enterprise-level indicators are as follows. (i) Equity concentration, measured by the percentage number of the first largest shareholder: the larger the percentage of

the first largest shareholder, the higher the equity concentration of the enterprise affects the role played by environmental regulation in improving the quality of environmental disclosure. (ii) Assets: asset size affects investors' investment decisions and enterprises' operating capacity and financial position. Specifically, the larger the enterprise, the easier it is to achieve scale effects. (iii) Operating revenue, expressed in terms of main business revenue: the higher the operating revenue, the higher the market share of the enterprise. (iv) Enterprise growth, expressed in terms of Tobin's Q, which is the ratio of market capitalization to total assets at the end of the period: the higher the Tobin's Q, the higher the market performance of the enterprise. (v) Marketing expenses, expressed by the marketing expense ratio, which is the ratio of marketing expenses to operating revenue: the higher the marketing expenses, the higher the cost to the business and the potential for some investment. (vi) Bank loan, measured by the ratio of the sum of short-term and long-term loans to total assets: bank loan requires a certain degree of creditworthiness; it reflects the enterprise's environmental information disclosure to a certain extent. (vii) Operating profit, expressed by net profit growth rate, namely the increase in the current net profit compared to the previous net profit: the profitability of an enterprise is reflected in the level of profit, and an enterprise with strong profitability can enhance the market's confidence, thus increasing its value. (viii) Return on assets, measured by the ratio of net profit to shareholders' average rights and interests: the higher the return on assets, the better the enterprise's performance and capital flow, allowing it to increase its allocation of more environmental funds.

The city-level indicators are as follows. (i) Innovation environment, expressed as the ratio of R&D investment to GDP: the greater the investment in R&D, the greater the incentive for enterprises to innovate. (ii) Economic development level, expressed as the ratio of GDP to permanent population: the higher the ratio, the better the economic development. (iii) Environmental supervision, expressed as the ratio of the number of provincial environmental inspection stations to administrative area: the greater the number of environmental inspection stations, reflecting the importance attached to environmental regulation, the stronger the policy implementation will be. The data are from the annual reports of listed companies, the Wind Information Database, the China City Statistical Yearbook, and the China Environmental Statistical Yearbook. Table 1 shows the descriptive statistical results of the variables.

**Table 1.** Descriptive statistical results.

| Variable | Sample | Label | Mean | Std. Dev | Min | Max |
|---|---|---|---|---|---|---|
| Active behavior | | AEB | 0.612 | 13.186 | 0.000 | 736.000 |
| Evasive behavior | | EAD | 1.898 | 2.773 | 0.000 | 12.430 |
| Environmental credit constraints | | ECC | 0.177 | 0.381 | 0.000 | 1.000 |
| Equity concentration | | TOP1 | 35.276 | 14.999 | 0.290 | 99.000 |
| Assets | | SIZE | 22.004 | 1.321 | 14.758 | 28.636 |
| Operating income | enterprise-level | SALE | 21.397 | 1.536 | 11.599 | 28.718 |
| Enterprise growth | | TOBIN | 2.130 | 2.481 | 0.153 | 126.952 |
| Marketing expenses | | SF | 0.071 | 0.094 | 0.000 | 4.843 |
| Bank loan | | BL | 0.170 | 0.169 | 0.000 | 10.689 |
| Operating profit | | OPR | −0.004 | 0.850 | −113.550 | 40.867 |
| Return on assets | | ROE | 0.063 | 5.529 | −186.557 | 713.204 |
| Innovation environment | | INE | 0.006 | 0.005 | 0.001 | 0.041 |
| Economic development | City-level | PGDP | 2.421 | 0.051 | 2.185 | 2.569 |
| Environmental supervision | | ES | 0.116 | 0.160 | 0.002 | 3.809 |

*4.2. Model Setting*

The environmental credit system can effectively constrain environmental behavior and curb enterprises' environmental violation problems and ecological damage, thus helping to achieve the goal of the construction of an ecological civilization. In this paper, we use data on China's A-share listed enterprises from 2009 to 2019 as the research sample

and the difference-in-differences (DID) model to test the impact of environmental credit constraints on enterprises' environmental behavior. The following two-way fixed effects model is constructed:

$$CEB_{it} = \alpha_0 + \alpha_1 lnECC_{it} + \sigma \text{ Control} + \theta_i + \mu_t + \epsilon_{it} \tag{1}$$

$\alpha_0$ is the intercept term, $\alpha_1$ is the net effect of the policy, $\sigma$ is the regression coefficient of the control variable, and Control denotes the set of control variables. $\theta_i$ and $\mu_t$ are enterprise and time fixed effects, respectively, and the two-way fixed effects model mitigates the omitted variable problem of the model. $\epsilon_{it}$ is the random error term, $i$ is the enterprise dimension, and $t$ is the time dimension.

## 5. Results

### 5.1. Benchmark Regression

The DID method was applied on the premise that there were no systematic differences between the target variables in the treatment and control groups before policy implementation, satisfying the parallel trend assumption. That is, in the absence of the exogenous impact of the blacklist system for illegal and dishonest environmental behavior, there is no systematic difference among enterprises' environmental protection behavior, and the two groups should maintain a common trend regarding changes. We take the year of policy implementation as the boundary, and the observations in each year of the treatment group before the policy implementation are sequentially assigned either negative or zero values, generating the variables ECC$_{-7}$, ..., ECC$_{-1}$. After the implementation of the policy, the samples for each year in the treatment group are sequentially assigned either positive or zero values, generating the variables ECC$_1$, ..., ECC$_3$. We assign a value of zero to the variable in the year the policy has been implemented. The parallel trend assumption requires that the regression coefficient of the variable be not significant before policy implementation, and whether it is significant after this implementation does not affect the prior conclusion. To present the regression results directly, the regression results of the variables after policy implementation were not presented. As seen from Table 2, before the implementation of the blacklist system, the impact of environmental credit constraints on enterprise environmental behavior was not significant, and there is no systematic difference between the treatment group and the control group, which meets requirements for parallel trends.

**Table 2.** Parallel trend test results.

| Variable | Active | Evasive |
|---|---|---|
| Year-7 | 0.212 (0.114) | −0.025 (0.014) |
| Year-6 | 0.197 (0.126) | −0.021 (0.016) |
| Year-5 | 0.218 (0.137) | 0.022 (0.017) |
| Year-4 | 0.246 (0.168) | 0.005 (0.020) |
| Year-3 | 0.235 (0.229) | 0.029 (0.027) |
| Year-2 | 0.178 (0.333) | 0.025 (0.041) |
| Year-1 | −0.234 (0.656) | 0.113 (0.083) |
| Cons | −3.508 (18.225) | 1.857 ** (0.779) |

**Table 2.** *Cont.*

| Variable | Active | Evasive |
|---|---|---|
| Control variables | Yes | Yes |
| Time fixed effect | Yes | Yes |
| Enterprise fixed effect | Yes | Yes |
| City fixed effect | Yes | Yes |
| Obs | 7325 | 16,372 |
| $R^2$ | 0.037 | 0.042 |

Note: ** indicates significance at 5%. Standard errors are reported in parentheses.

As shown in Table 3, environmental credit constraints can not only strengthen enterprises' AEB but also produce EAD. The regression results of AEB show that environmental credit constraints have a positive impact on enterprises' environmental protection behavior under a 5% significance level, and the regression coefficient is 1.143, indicating that the construction of the blacklist system of fraudulent and dishonest environmental behavior is helpful in promoting enterprises' green technology innovation to improve their ability to control pollution and their credit. The regression coefficient of EAD is 0.126, which is significant at the 5% level, indicating that environmental credit constraints increase the number of dismissals and resignations. This will reduce enterprises' production and have a negative impact on their operating conditions, thus restraining the growth of their long-term business performance. The results show that the blacklist system of environmental fraud is unfavorable to enterprises adopting EAD, while the active response strategy is the long-term solution. Environmental credit constraints are conducive to generating the driving force of innovation in environmental protection and can encourage enterprises to carry out green technology innovation [52], resulting in production activities that are more efficient and sustainable. However, the environmental and economic pressure brought by environmental credit constraints will also increase the production costs of enterprises [53], including environmental taxes, pollution control equipment inputs, and environmental confiscation costs, causing some enterprises to choose to evade environmental protection behavior to relieve the pressure of environmental regulations, which supports H1.

**Table 3.** Benchmark regression results.

| Variable | Active | Evasive |
|---|---|---|
| lnECC | 1.143 ** | 0.126 ** |
| | (0.486) | (0.053) |
| Cons | −12.179 | −1.705 |
| | (19.309) | (2.615) |
| Control variables | Yes | Yes |
| Time fixed effect | Yes | Yes |
| Enterprise fixed effect | Yes | Yes |
| City fixed effect | Yes | Yes |
| Obs | 7305 | 16,354 |
| $R^2$ | 0.038 | 0.046 |

Note: ** indicates significance at 5%. Standard errors are reported in parentheses.

*5.2. Robustness Test*

(1) PSM-DID

In the real economy, some pilot policies may be nonrandom (or quasi-natural experiments), and using the DID model may result in self-selection bias for policy effect assessments. Propensity score matching (PSM) can match the specific control group samples for the treatment group samples, overcome the endogenous influence of sample selection, and make the quasi-natural experiment approximately random. The PSM-DID method was adopted to further test the influence of environmental credit constraints on enterprise environmental protection behavior and identify the net effect of the blacklist system.

In the regression model, there may be problems with data bias and confounding variables, and the PSM method can reduce the impact of these issues on the measurement model to allow for a more systematic comparison between the treatment and control groups. PSM matching methods include one-to-one and one-to-many matching. One-to-one matching may cause a large number of samples to be lost, resulting in large variances and less deviations; one-to-many matching may cause fewer samples to be lost, which can reduce the variance but lead to large deviations. According to the research objectives, one-to-four return-back matching was adopted in this paper to minimize the mean square error. As a nonparametric method, PSM does not need to assume the conditional mean function of observable factors and the probability distribution of unobservable factors and thus has greater advantages over the parametric method. In this paper, assets, operating income, enterprise growth, and equity concentration were selected as matching variables for sample matching using the logit model. Table 4 shows the regression results of the PSM-DID model, and the influence coefficients of environmental credit constraints on active and evasive environmental protection behaviors are 1.143 and 0.126, respectively—both are significant at the 5% level. The conclusions of the study are basically consistent with those of the benchmark regression, and the results are robust.

**Table 4.** Regression results of PSM-DID model.

| Variable | Active | Evasive |
|---|---|---|
| lnECC | 1.107 ** | 0.150 ** |
| | (0.481) | (0.052) |
| Cons | −10.615 | −0.564 |
| | (19.718) | (2.650) |
| Control variables | Yes | Yes |
| Time fixed effect | Yes | Yes |
| Enterprise fixed effect | Yes | Yes |
| City fixed effect | Yes | Yes |
| Obs | 6833 | 15,362 |
| $R^2$ | 0.036 | 0.045 |

Note: ** indicates significance at 5%. Standard errors are reported in parentheses.

The PSM method needs to meet the balance and common support assumptions to ensure the validity of the matching results. First, the balance assumption test. The purpose of this test is to observe whether there is a significant difference in the matched covariates between the treatment and control groups, and if so, whether a good matching effect is indicated. There are two main balance test methods: one is to measure the deviation in the standardized mean of the covariate in the two groups, and the other is to judge whether the value of each covariate is systematically different between the two groups through the P test. The second test was used in this paper. According to Table 5, the *p* value changes from significant to insignificant from before to after matching. The sample deviation after matching was greatly reduced, and there were no systematic differences between the treatment and control groups, showing that the PSM results passed the balance assumption test.

**Table 5.** Balanced hypothesis test results.

| Variables | Sample | Active | | Evasive | |
|---|---|---|---|---|---|
| | | Sample Bias | *p*-Value | Sample Bias | *p*-Value |
| SIZE | Before matching | −4.700 | 0.019 | −18.900 | 0.000 |
| | After matching | 0.800 | 0.715 | 1.000 | 0.514 |
| SALE | Before matching | 5.700 | 0.004 | −11.400 | 0.000 |
| | After matching | 0.700 | 0.724 | 1.300 | 0.427 |
| TOP1 | Before matching | | | −10.600 | 0.000 |
| | After matching | | | −0.300 | 0.841 |
| TBQ | Before matching | | | 2.800 | 0.059 |
| | After matching | | | −1.800 | 0.248 |

Second, the co-support assumption test. The co-support assumption is also known as the overlap assumption, which requires the propensity score values of the treatment and control groups to overlap with enough areas to improve the matching quality by retaining only the individuals of the overlapping areas. In this paper, the kernel density function of the two sample groups before and after matching was drawn to visually observe the matching effect. Figures 1 and 2 are the estimated results of the kernel density function before and after matching when adopting the enterprise's AEB as the explained variable. In Figures 3 and 4, the explained variable is enterprise EAD. The overlap degree between the matched treatment and control groups is relatively high, and the changes show a significant convergent trend. The fitting degree is obviously better than the results before matching, and the samples meet the assumption of co-support.

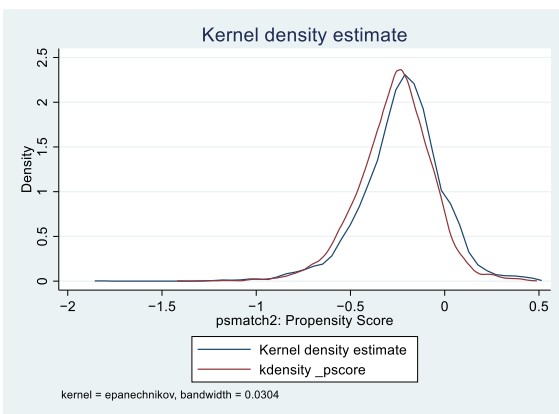

**Figure 1.** Before matching (active type).

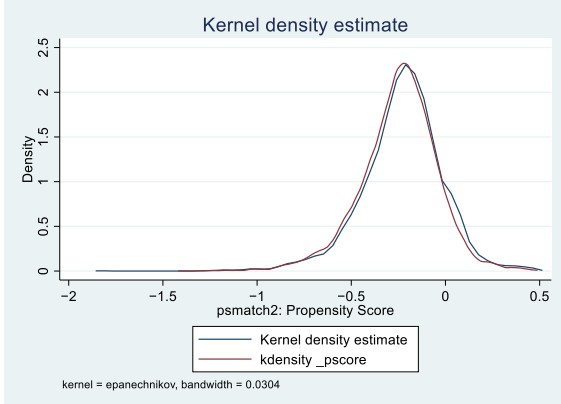

**Figure 2.** After matching (active type).

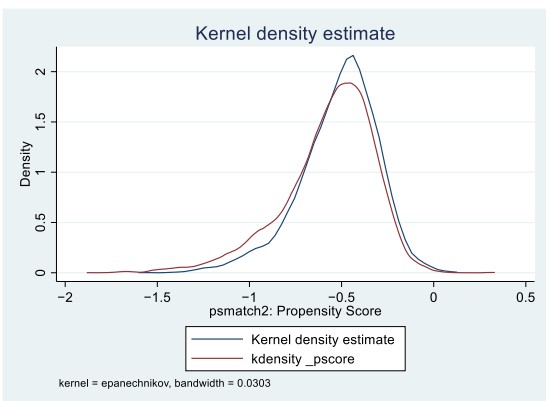

**Figure 3.** Before matching (evasive type).

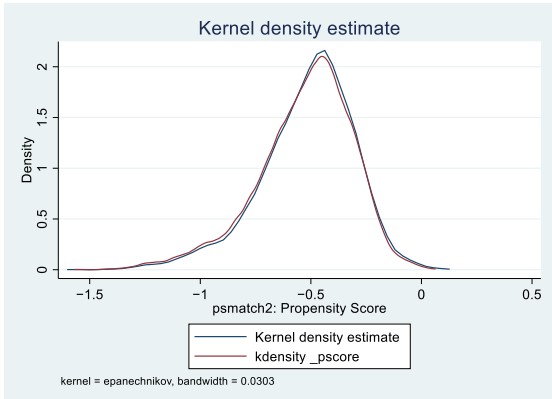

**Figure 4.** After matching (evasive type).

(2) Placebo test of the treatment group

Although the model met the parallel trend requirements, it was unknown whether the changes in the trend in the treatment and control groups were influenced by other randomness factors after policy intervention. It was difficult to determine whether the net effect of policies in the model was the influence of environmental credit constraints on enterprises' environmental protection behavior. To ensure the robustness of the regression results, a placebo test was performed on the treatment group, excluding the impact on the samples by other factors in the same period as much as possible. The following two methods were adopted to process the treatment group. First, a treatment group was fabricated. We sorted the enterprise codes and selected the same number of samples as in the treatment group to produce a new treatment group and multiplied it by the time variable to form a new explained variable. According to placebo test 1 in Table 6, the regression coefficients for both active and evasive environmental protection behavior are not significant, meaning that for this fictitious treatment group, there are no concurrent factor-driven results, so the previous conclusions are robust. Second, areas with higher attention to environmental quality and environmental regulation intensity may take the lead in implementing the blacklist system of environmental fraud and dishonesty [54], which leads to the nonrandom selection of treatment groups. To exclude the above situation, we ranked the intensity of the environmental regulations (i.e., the proportion of environmental pollution control investment in GDP) and selected the same number of samples as in the treatment group to produce a new treatment group. According to placebo test 2 in Table 6, the regression coefficients of both the active and evasive environmental protection behaviors are not significant, which can exclude the interference of environmental regulations on the results, and the benchmark regression results are robust.

**Table 6.** Placebo test results.

| Variable | Placebo Test 1 | | Placebo Test 2 | | Placebo Test 3 | |
|---|---|---|---|---|---|---|
| | **Active** | **Evasive** | **Active** | **Evasive** | **Active** | **Evasive** |
| lnECC | 0.035 | −0.781 | −0.384 | −0.061 | −0.012 | −0.061 |
| | (5.099) | (0.807) | (0.405) | (0.042) | (0.115) | (0.042) |
| Cons | −7.759 | −1.478 | −6.778 | −1.796 | −7.647 | −1.796 |
| | (19.319) | (2.618) | (19.199) | (2.616) | (19.194) | (2.616) |
| Control variables | Yes | Yes | Yes | Yes | Yes | Yes |
| Time fixed effect | Yes | Yes | Yes | Yes | Yes | Yes |
| Enterprise fixed effect | Yes | Yes | Yes | Yes | Yes | Yes |
| City fixed effect | Yes | Yes | Yes | Yes | Yes | Yes |
| Obs | 7325 | 16,372 | 7325 | 16,372 | 7325 | 16,372 |
| $R^2$ | 0.029 | 0.046 | 0.031 | 0.046 | 0.029 | 0.046 |

(3) Placebo test of the policy implementation time

To ensure the robustness of the regression results, counterfactual tests can be conducted by changing the policy implementation time. In the model, the net effect of policies may not be entirely generated by the blacklist system of environmental fraud and dishonesty, and other policies or random factors may also affect enterprises' environmental protection behavior in the same period. Furthermore, some enterprises may predict the implementation time of the policy and make pollution reduction and innovation reactions in advance, which leads to bias in the regression results. To exclude the influence of such factors, we advanced the policy implementation time by two years to produce new policy variables. A significantly positive regression coefficient of the variables indicates that the choice of enterprises' environmental protection behavior may be influenced by other policies or random factors. If the regression coefficient is not significant, then there is no systematic difference in the trend in the changes in the environmental protection behavior between the treatment and control groups after excluding the impact of the blacklist system of environmental fraud and dishonesty. According to placebo test 3 in Table 6, the regression coefficients of both active and evasive environmental behaviors are not significant, which means that the net effect of the policy is generated by the blacklist system.

(4) Add other policy variables

The robustness of the regression results was further tested by adding policy variables, including the carbon emission rights trading policy and the environmental protection supervision resident system. First, the carbon emission rights trading policy will affect enterprises' environmental protection behavior [55]. Before the implementation of this policy, enterprises reduced carbon emissions mainly to fulfill their social responsibilities. However, after the implementation of this policy, enterprises can trade the surplus emission rights quota, and the greater the balance is, the greater the economic benefits that enterprises obtain. For some enterprises with excess emissions, the carbon emission rights trading policy can encourage them to increase investments in carbon emissions reductions and accelerate the upgrading of product quality and production mode transformation through technological innovation to reduce environmental costs. Second, the environmental protection supervision resident system enables the central government to dispatch environmental protection supervision agencies to some provinces to be responsible for ecological and environmental protection supervision in the resident and surrounding areas. This system effectively constrains the environmental pollution control behavior of local governments and enterprises and reduces the frequency of illegal environmental activities. As seen from Table 7, after adding other policy variables, the regression coefficient of environmental credit constraints is significantly positive for both active and evasive environmental protection behaviors, which indicates that the benchmark regression results are robust.

**Table 7.** Results after adding other policy variables.

| Variable | Active | Evasive |
|---|---|---|
| lnECC | 0.818 * | 0.146 *** |
| | (0.500) | (0.054) |
| Cons | −19.669 | −1.616 |
| | (19.482) | (2.624) |
| Control variables | Yes | Yes |
| Time fixed effect | Yes | Yes |
| Enterprise fixed effect | Yes | Yes |
| City fixed effect | Yes | Yes |
| Obs | 7325 | 16,372 |
| $R^2$ | 0.050 | 0.047 |

Note: *** and * indicate significance at 1 and 10%, respectively. Standard errors are reported in parentheses.

### 5.3. Heterogeneity Analysis

(1) Property rights attributes

Property rights are one of the main factors affecting environmental protection behavior. Enterprises with different property rights have different attitudes toward undertaking social responsibility and protecting the environment, leading to different environmental protection strategies. According to the different nature of the controlling shareholder and the actual controller, enterprises can be divided into state-owned and private enterprises, and the two types of enterprises have differentiated responses to the government's environmental protection policies. State-owned enterprises emitting high pollution are the key monitoring object of the environmental protection department [56] and are subject to stricter supervision over pollution control, environmental monitoring, and project approvals, etc. Moreover, the government background of state-owned enterprises enables them to undertake more policy functions, and the will of local governments is often reflected more in their investment and financing, innovation, and environmental protection behaviors. While pursuing economic benefits, these enterprises also give more consideration to their social responsibilities. In addition, with the continuous promotion of the national ecological civilization strategy, environmental protection indicators have gradually been included in the assessment system of government officials and senior executives of state-owned enterprises, making state-owned enterprises pay more attention to environmental pollution problems to gain greater promotion advantages. Private enterprises do not have the advantages and ability of state-owned enterprises to obtain resources, and their primary goal is to maximize shareholder interests [57]. Given environmental protection investments with high investment costs and long capital recovery periods, private enterprises are more inclined to allocate resources to fields with high economic benefits.

As shown in Table 8, environmental credit constraints encourage private enterprises to choose the evasive environmental protection strategy, while state-owned enterprises may choose the active environmental protection strategy. The influence coefficient of environmental credit constraints on the AEB of state-owned enterprises is 0.023, which is significant at the 5% level, while the influence on evasive environmental behavior is not significant. In comparison, private enterprises have made the opposite choice mainly because technological innovation is generally characterized by high investments [58], long cycles, low short-term profits, high risk, and high trial and error costs, leading private enterprises to lack enthusiasm for innovation. Therefore, when faced with environmental credit constraints, private enterprises are more inclined to avoid environmental penalties by downsizing, reducing production, and other ways. State-owned enterprises have diversified financing channels and institutional guarantees, which can provide continuous financial support for technological innovation. In addition, state-owned enterprises are not owned by individuals but are public property enterprises, and staffing is fixed. Layoffs may stimulate the contradiction between management and employees, bring more uncertainty to enterprise development, and affect the promotion of management personnel. Therefore,

when faced with environmental credit constraints, state-owned enterprises may choose the active environmental protection strategy.

**Table 8.** Regression results for differences in property rights.

| Variable | State-Owned Enterprises | | Private Enterprises | |
|---|---|---|---|---|
| | **Active** | **Evasive** | **Active** | **Evasive** |
| lnECC | 0.023 ** | 0.245 | 2.002 | 0.023 ** |
| | (0.079) | (0.107) | (0.854) | (0.053) |
| Cons | 1.579 | 0.547 | −29.282 | 0.333 |
| | (2.956) | (4.405) | (34.657) | (2.713) |
| Control variables | Yes | Yes | Yes | Yes |
| Time fixed effect | Yes | Yes | Yes | Yes |
| Enterprise fixed effect | Yes | Yes | Yes | Yes |
| City fixed effect | Yes | Yes | Yes | Yes |
| Obs | 2817 | 7008 | 4154 | 8331 |
| $R^2$ | 0.037 | 0.047 | 0.062 | 0.079 |

Note: ** indicates significance at 5%. Standard errors are reported in parentheses.

(2) External interest association

Stakeholders' reasonable economic and environmental demands directly affect enterprises' production and operating activities, and the difference in the degree of correlation and expected goals among stakeholders may force enterprises to adopt differentiated environmental protection strategies. The impact of external interest associations was mainly discussed in this paper, with external stakeholders mainly including shareholders, suppliers, and creditors. China's environmental protection system is becoming increasingly mature, environmental taxes have had an important impact on enterprises, and the public awareness of environmental protection has also been improving. Given the combined action of these factors, shareholders may pursue economic interests, while taking into account more environmental responsibilities in decision making, and guide enterprises to actively consider environmental risks and returns when making investment decisions. However, at the same time, if enterprise stakeholders pay more attention to economic benefits and ignore environmental benefits, enterprises may produce more environmental fraud or adopt more negative environmental protection strategies and weaken the green development effect of environmental credit constraints. In this paper, grouping was based on whether there was an interest association between the top ten shareholders, that is, the results are for a high interest and a low interest correlation group.

As seen from Table 9, when faced with environmental credit constraints, enterprises with high interest correlations tend to choose AEBs, while enterprises with low interest correlations tend to adopt evasive environmental protection strategies. When the correlation of the external interests of enterprises is high, the regression coefficient of environmental credit constraints affecting AEB is 1.810, which is significant at the 5% level, while the impact on EAD is not significant. When the correlation of external interests of the enterprise is low, the environmental credit constraints make the enterprise pursue EADs, and the regression coefficient is 0.338, which is significant at the 1% level. There may be to reason behind this phenomenon. First, enterprises with higher external interest correlation may be disturbed by more human factors, such as regarding employee recruitment, job divisions, rights arrangements, and other aspects. Therefore, when environmental credit constraints have a negative impact on enterprise production activities, the enterprise decision makers minimize their consideration of dismissing employees or reducing production costs to avoid conflicts with other stakeholders. Second, the blacklist system of environmental fraud and dishonesty increases the risk and cost of enterprise environmental fraud and guides decision makers to make decisions conducive to environmental protection. There is a lower degree of information asymmetry between the stakeholders of enterprises with higher interest correlation [59]. Shareholders can have a more comprehensive understanding of

enterprise environmental information disclosures, which can show a good corporate image to the market and reduce investors' uncertainty regarding the prospects of the enterprise to improve its market value, thus encouraging enterprises to adopt an active environmental protection strategy to form positive interactions with the market's active responses.

**Table 9.** Regression results for differences in external interest association.

| Variable | High Interest Association | | Low Interest Association | |
| --- | --- | --- | --- | --- |
| | **Active** | **Evasive** | **Active** | **Evasive** |
| lnECC | 1.810 ** | 0.059 | 0.075 | 0.338 *** |
| | (0.884) | (0.064) | (0.078) | (0.098) |
| Cons | −27.692 | 5.817 * | −0.586 | −17.562 *** |
| | (37.423) | (3.301) | (2.746) | (3.956) |
| Control variables | Yes | Yes | Yes | Yes |
| Time fixed effect | Yes | Yes | Yes | Yes |
| Enterprise fixed effect | Yes | Yes | Yes | Yes |
| City fixed effect | Yes | Yes | Yes | Yes |
| Obs | 3879 | 8722 | 3446 | 7650 |
| R$^2$ | 0.076 | 0.051 | 0.074 | 0.054 |

Note: ***, ** and * indicate significance at 1, 5 and 10%, respectively. Standard errors are reported in parentheses.

(3) Enterprise profitability

Enterprises' environmental protection investments increase their management and production costs. Although the blacklist system of environmental fraud and dishonesty can restrict enterprises' environmental protection behavior, enterprises ultimately aim to pursue profit maximization. Only when their profitability reaches a certain level do they have a greater ability to address environmental problems. The return on assets is the ratio of profits to average assets, which is an index that reflects the comprehensive utilization effect of assets. The higher the return on assets, the higher the utilization efficiency of assets, the stronger the profitability of the enterprise, and the higher the level of operations and management, which can achieve good results in increasing income and saving funds [60]. Therefore, return on assets reflects the level of enterprise management and the implementation of the system of responsibilities. A decrease in return on assets indicates that the enterprise's profits have declined, and its production and operating capacity has been greatly affected. Return on assets was selected to show enterprise profitability, and the sample is divided into two groups using the mean value of return on assets.

As seen from Table 10, high-profitability enterprises choose active environmental protection strategies when facing environmental credit constraints, while low-profitability enterprises tend to choose evasive environmental protection strategies. For high-profit enterprises, the influence coefficient of environmental credit constraints on AEB is 2.738, which is significant at the 1% level, while the impact on EAD is not significant. In comparison, low-profitability companies have the opposite results because in the case of higher return on assets, the enterprise's management has a stronger motivation to voluntarily disclose environmental information to maintain the sustainability of its position and its remuneration. The disclosure of environmental information means that enterprises can better identify and evaluate environmental issues [61] and, on this basis, take the correct and sustainable AEB. At this time, such enterprises increase their investments in environmental protection and promote green technology innovation to improve their environmental performance. The reason enterprises with a low return on assets are more inclined to EAD lies in the high cost of technological innovation. Such enterprises do not have enough capital investments to ensure the smooth progress of technology research and development, and due to the influence of strict environmental protection policies, the production cost of unit products increases; as a result, they may cope with environmental credit constraints through layoffs and reducing production.

**Table 10.** Regression results for difference in profitability.

| Variable | High Profitability | | Low Profitability | |
|---|---|---|---|---|
| | **Active** | **Evasive** | **Active** | **Evasive** |
| lnECC | 2.738 *** | 0.040 | −0.115 | 0.335 *** |
| | (1.096) | (0.076) | (0.076) | (0.082) |
| Cons | −28.775 | 13.973 ** | 2.545 | −3.526 |
| | (41.512) | (6.003) | (2.983) | (3.816) |
| Control variables | Yes | Yes | Yes | Yes |
| Time fixed effect | Yes | Yes | Yes | Yes |
| Enterprise fixed effect | Yes | Yes | Yes | Yes |
| City fixed effect | Yes | Yes | Yes | Yes |
| Obs | 3800 | 8546 | 3525 | 7826 |
| $R^2$ | 0.077 | 0.063 | 0.044 | 0.043 |

Note: *** and ** indicate significance at 1 and 5%, respectively. Standard errors are reported in parentheses.

## 6. Working Channels and Further Discussion

### 6.1. Working Channels

(1) R&D investments

R&D investments are important to promoting green technology innovation. Reports disclosed by enterprises show that R&D investments are mainly used in two aspects, including R&D personnel investments and R&D fund investments. First, from the perspective of R&D personnel investments, R&D personnel are the main carriers of enterprises' green technology innovation and the key to achieving their goal of green technology innovation. R&D personnel with high education, high knowledge, and rich experiences usually have a higher technical level and stronger innovation consciousness, which can greatly promote enterprises' green technology innovation. The more investment in R&D personnel, the larger the number of highly skilled personnel that can be recruited. Given sufficient R&D funds, the skills of the R&D personnel and enterprises' ability to engage in green technology innovation are significantly improved. Second, R&D investments have far-reaching significance for green technology innovation activities. When R&D investments continue to increase, the enterprise's R&D facilities become increasingly advanced and are better able to provide employees with a better scientific research environment and create a strong scientific research atmosphere. Therefore, the more sufficient R&D funds there are, the more capable enterprises are to continue to carry out green technology innovation activities. Thus, R&D investments have a positive impact on AEB.

How do environmental credit constraints affect R&D investments? As shown in Table 11, a regression coefficient of 0.343 that is significant at the 5% level shows that environmental credit constraints encourage enterprises to increase their R&D investments. This shows that the stronger the environmental credit constraints are, the higher enterprise R&D investments are, and environmental credit constraints have a significant positive effect on R&D investments. Environmental credit constraints represent the performance of the integrity of enterprise environmental protection and a good opportunity to shape enterprises' culture of internal integrity. Environmental credit constraints form a strong deterrent to enterprises' environmental damage through the compulsory measures of "a breach of trust, everywhere restricted"; therefore, enterprises under the pressure of environmental credit constraints increase their environmental protection investments, which supports H2. This incentive effect makes up for the increase in production costs resulting from environmental policies, namely the innovation compensation effect, enabling related production activities to increase long-term economic benefits while remaining in line with the requirements of the environmental system. Additionally, the flexible supervision of credit by government departments provides more convenient services for enterprises that keep their word, including credit incentives and innovation subsidies, which strengthen enterprises' motivation for environmental protection, encourage them to continuously increase R&D investments, and improve their green technology innovations.

**Table 11.** Mechanism test results.

| Variable | RDIM | FEC |
|---|---|---|
| lnECC | 0.343 ** | 0.008 ** |
|  | (0.157) | (0.003) |
| Cons | 25.642 *** | 0.066 |
|  | (3.068) | (0.165) |
| Control variables | Yes | Yes |
| Time fixed effect | Yes | Yes |
| Enterprise fixed effect | Yes | Yes |
| City fixed effect | Yes | Yes |
| Obs | 14,076 | 18,652 |
| $R^2$ | 0.088 | 0.085 |

Note: *** and ** indicate significance at 1 and 5%, respectively. Standard errors are reported in parentheses.

(2) Debt size

An enterprise's amount of debt is closely related to its production scale and business activities. When an enterprise's debt reaches a certain scale, it reduces economic activities in other ways to save costs and address the pressure from the debt. On the one hand, high debt could make enterprises more vulnerable to financial crises, forcing them to take austerity measures to reduce costs—layoffs are often one of the simplest and most straightforward cost-cutting measures considered. The main reason that layoffs can reduce production costs is that they reduce the human resource costs of enterprises. Layoffs include separation, social security, and provident fund costs, which are limited relative to the cost of employing an employee for a long period and the cost of reducing production. Furthermore, the exemplary effect of layoffs may help enterprises reduce the wages of other workers, thus further reducing the pressure they face. In addition to human resource costs, layoffs can reduce other production costs, such as equipment maintenance and material supplies. When production is reduced, enterprises can reduce the production equipment or materials used, also reducing production costs and resulting in a lower amount of debt that enterprises need to repay. On the other hand, the target of layoffs is generally a low-skilled, low-efficiency labor force. Reducing the proportion of these workers can improve an enterprise's production efficiency and profit margins, thus increasing its cash flow and solvency.

According to Table 11, environmental credit constraints significantly increase the scale of an enterprise's debt. The regression coefficient is 0.008, which is significant at the 5% level. This is because environmental credit constraints impose new constraints on enterprises' production decisions and make them pay more attention to green products and product quality, leading to more difficulties in the links among management, production, and sales, thus increasing production and operation costs. Given these conditions, enterprises are likely to scale up their debt to obtain more financial support. Environmental credit constraints reflect enterprises' disclosures of environmental fraud based on environmental protection policy requirements, especially related to the negative impact of enterprise production and operations on the environment. Although the disclosure of this type of information reduces the information asymmetry for market investors, it undoubtedly sends a signal that the enterprise has failed to pay attention to environmental protection, exposes its environmental risks, and creates concerns for investors, adversely impacting enterprise financing. Given the profit-driven induction of capital, investors turn to other enterprises. From the perspective of banks, in the context of the high-quality development and construction of an ecological civilization, banks pay more attention to enterprises' environmental information and environmental credit when making investment decisions and screen enterprises with high environmental credit and large financing needs to provide more financial support. In contrast, environmental trust-breaking enterprises face more difficulties obtaining financial support and, thus, experience excessive growth in debt, which supports H3.

### *6.2. Further Discussion*

At the present stage, some enterprises still lack the necessary awareness of labor security and are prone to labor conflicts caused by infringing on workers' rights and interests. The production and operating goal of an enterprise is to maximize profits, and one way to achieve this goal is to reduce production costs, such as workers' wages, which damage the legitimate rights and interests of workers and bury the deep hidden danger of labor conflicts. Some enterprises lay off workers when faced with a number of difficulties in their production and operating activities that force them to reduce production. At this time, the following three situations may occur. First, some employees are about to reach retirement age, and enterprises may encourage them to apply for early retirement. Second, due to the enterprise production scale, the supply of low-skilled workers exceeds its demand, and enterprises reduce labor costs through layoffs. However, since these employees are protected by contract law, if the two parties cannot reach an agreement on the changes in the labor contract through negotiations, dismissed employees apply for labor arbitration or bring lawsuits. Third, some managers with high salaries and strong job substitution are also targets of layoffs when enterprises face substantial negative situations. However, these employees are strongly aware of their rights and have high interest demands; therefore, they are more likely to engage in labor arbitration and file lawsuits against enterprises. Thus, it can be seen that when enterprises choose the evasive environmental protection strategy, one of the potential hidden dangers is labor conflicts, which affect enterprises' reputations and production activities. Will environmental credit constraints increase enterprise labor conflict? In this paper, labor conflicts are measured by litigation and arbitration cases to analyze whether EAD caused by environmental credit constraints aggravates labor conflicts.

As shown in Table 12, EAD caused by environmental credit constraints increases the probability of litigation and arbitration cases. After adding the control variable, the regression coefficient of the interaction term between environmental credit constraints and EAD was 0.006 and significant at the 10% level. Environmental credit constraints have a substantial impact on the production and operating activities of heavily polluting enterprises, forcing some enterprises to choose evasive environmental protection strategies to reduce the environmental cost caused by environmental regulations. Such enterprises may directly transfer environmental costs to workers, reduce wages, or lay off workers, affecting the stability of labor–capital interests and aggravating labor–capital conflicts. Thus, it can be seen that when implementing the blacklist system of environmental fraud and dishonesty, the government should consider not only the effect of ecological environmental protection but also the affordability of enterprises to prevent them from transferring the costs to workers and damaging worker interests. In the short term, the construction of an environmental credit system may aggravate unemployment, especially through an increase in the short-term unemployment rate of high-pollution and high-cost enterprises; therefore, the government must pay attention to these issues when formulating and implementing policies.

**Table 12.** Regression results of litigation and arbitration.

| Variable | LAWS | LAWS |
|---|---|---|
| lnECC × lnEAD | 0.007 ** | 0.006 * |
| | (0.003) | (0.003) |
| Cons | 0.229 | 0.352 ** |
| | (0.164) | (0.172) |
| Control variables | No | Yes |
| Time fixed effect | Yes | Yes |
| Enterprise fixed effect | Yes | Yes |
| City fixed effect | Yes | Yes |
| Obs | 20,023 | 18,113 |
| $R^2$ | 0.144 | 0.147 |

Note: ** and * indicate significance at 5 and 10%, respectively. Standard errors are reported in parentheses.

## 7. Conclusions and Recommendations

The blacklist system of environmental fraud enables the government to make environmental information public in the form of environmental policies, establish a joint disciplinary mechanism for breaches of trust, and increase the cost of environmental fraud by changing the distribution structure of enterprises' capital and other factors. Based on a theoretical explanation of how environmental credit constraints affect the choices that enterprises make regarding their environmental protection behavior, we use the DID method to identify the effects of environmental credit constraints on active and evasive environmental protection behaviors using data on A-share listed companies in China. The results are as follows. (1) Environmental credit constraints prompt some enterprises to choose active environmental behaviors because of the incentive effect of the environmental credit constraint on R&D investments. However, some enterprises adopt evasive environmental strategies when faced with environmental credit constraints because of increased production costs and debt. (2) State-owned enterprises prefer active environmental strategies to address environmental credit constraints, while private enterprises adopt evasive strategies. Environmental credit constraints make enterprises with strong interest associations and high profitability choose active environmental strategies. Instead, they choose evasive strategies. (3) Environmental credit constraints generate evasive corporate environmental behavior that increases the probability of litigation and arbitration cases, and the construction of an environmental credit system may exacerbate unemployment in the short term, which is something the government needs to be aware of when developing and implementing a blacklist system of environmental fraud.

To further improve the environmental management system, strengthen the positive role of environmental credit system construction on pollution control, and explore effective environmental protection strategies suitable for different types of enterprises under the constraints of the environmental credit system, this paper proposes the following suggestions:

(1) The government needs to improve the blacklist system of environmental fraud, improve the incentive mechanism for environmental compliance, and further improve the economic benefits of environmental compliance for enterprises. The government can promote the development of green finance, such as green credit, green bonds, and green insurance, encourage and guide social capital to support environmentally trustworthy enterprises, especially private enterprises, establish a diversified ecological environmental protection investment and financing mechanism for private enterprises, and provide continuous financial support for their green technology innovation. Such actions motivate enterprises to comply with the environmental credit system and carry out green innovation activities.

(2) The innovation environment requires optimization and innovation costs to be reduced. The high risk and high cost of technology innovation have inhibited the motivation of some enterprises to engage in technology innovation. Therefore, all sectors of society need to work together to create a social culture and effective market competition environment that is conducive to enterprises' independent innovation, reduce the cost of maintaining intellectual property rights, and strengthen the effective protection of intellectual property rights. The innovation system needs to be improved at different levels, the investment and supply of common technologies needs to be increased, priority needs to be given to supporting green innovation in strategic emerging industries, and green technology innovation platforms need to be built in different categories to enhance their support for enterprises' green development.

(3) According to the nature of the industry and the pollution situation, the blacklist system of environmental fraud strongly promoted by industry should seek a balance between economic growth and resource and environmental constraints, determine a rectification period for enterprises engaged in environmental violations, and exclude them from the environmental breach of trust list if they achieve predetermined environmental goals within the rectification period, thus reducing the negative impact on enterprises' production activities while achieving the pollution reduction goals of environmental credit constraints and avoiding social problems, such as an increase in short-term unemployment.

The limitations of this paper are as follows. The first is due to the limitations of the sample; the relevant data of small- and medium-sized enterprises are difficult to obtain, and we have to use the data of listed companies, so the influence of small- and medium-sized enterprises have not been considered in the research question, which may make the sample have a certain bias. Second, for the limitation of the research object, this paper only aims to study active environmental protection behavior and evasive environmental protection behavior and does not consider the situation of passive environmental protection behavior, which may lead to a conclusion with certain limitations.

Future research can be further carried out in the following aspects: First, expand the research scope to longer periods and then observe whether the findings of this paper are still valid. Second, in subsequent studies, data from small- and medium-sized enterprises can be added to the study to increase the sample size and sample range, so as to reduce sample bias. Third, further research is needed on the impact of environmental credit constraints on enterprises' environmental investments and other aspects.

**Author Contributions:** Methodology, G.L.; Formal analysis, X.X.; Investigation, X.X.; Data curation, G.L.; Writing—original draft, C.Y.; Writing—review & editing, L.L. All authors have read and agreed to the published version of the manuscript.

**Funding:** This research was funded by Higher Education Research Subject of China Association of Higher Education, grant number 23BR0104; China-CEE Joint Education Program, grant number 2022195.

**Institutional Review Board Statement:** Not applicable.

**Informed Consent Statement:** Not applicable.

**Data Availability Statement:** The data presented in this study are available on request from the corresponding author.

**Conflicts of Interest:** The authors declare no conflict of interest.

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
