# Peer review of "Environmental Credit Constraints and the Enterprise Choice of Environmental Protection Behavior"

_sustainability, doi:10.3390/su152416638_

Round 1
Reviewer 1 Report
Comments and Suggestions for Authors
Thank you for letting me review this manuscript. The authors shed light on a burning problem, especially in the view that China emits more greenhouse gases than the entire developed world combined. The research by Rhodium Group says China emitted 27% of the world's greenhouse gases in 2019. Therefore, understanding how the Chinese government and companies respond to these challenges is crucial.
The abstract and the introduction are well-designed, they present the issue and point out the gap in research. The literature review is appropriate, covers current sources, and comprehensively covers the field, which is later treated in the empirical part. It also represents a good basis from which the three hypotheses are derived.
The methodology is well-grounded and thoroughly applied to data processing. The results are also well and comprehensibly presented, perhaps in the discussion the results could be somewhat connected with references and secondary literature.
Comments on the Quality of English LanguageI have no comments on the English language of the article. Maybe in line 79, "analyzes" could lose the final "s".
Author Response
Reviewer 1
Thank you for letting me review this manuscript. The authors shed light on a burning problem, especially in the view that China emits more greenhouse gases than the entire developed world combined. The research by Rhodium Group says China emitted 27% of the world's greenhouse gases in 2019. Therefore, understanding how the Chinese government and companies respond to these challenges is crucial.
The abstract and the introduction are well-designed, they present the issue and point out the gap in research. The literature review is appropriate, covers current sources, and comprehensively covers the field, which is later treated in the empirical part. It also represents a good basis from which the three hypotheses are derived.
The methodology is well-grounded and thoroughly applied to data processing. The results are also well and comprehensibly presented, perhaps in the discussion the results could be somewhat connected with references and secondary literature.
- I have no comments on the English language of the article. Maybe in line 79, "analyzes" could lose the final "s".
We would like to thank the reviewers for their positive feedback.
Comment 1: Maybe in line 79, "analyzes" could lose the final "s".
Many thanks for pointing this out. We have carefully checked the word for grammatical problems, deleted "s" from "analyses".
Reviewer 2 Report
Comments and Suggestions for Authors
1. Abstract: improve the consistency between abstract-introduction-problem definition- results-conclusions
Suggestion: include your main hypothesis or research question starting from abstract, research methods and research limitations. A short description including the relationship between variables would be a great explanation for the reader.
2. Lines 74, 75 and 76: Reference at least one study.
3. Lines from 159 to 171 seem to be some conclusions! If this is the case, please indicate that, e.g. by starting the paragraph with. In conclusion to this part...
In the case this is not the author's conclusion, please reference it.
4. Hypothesis 2 seems to consist also of this variable: 'green technology innovation'. It is not treated as variable. It is clear that this does not affect the results, however, a short explanation could help clarify the reason for its existence within hypothesis 2.
Well, interesting research work!
Author Response
Reviewer 2
- Abstract: improve the consistency between abstract-introduction-problem definition- results-conclusions
Suggestion: include your main hypothesis or research question starting from abstract, research methods and research limitations. A short description including the relationship between variables would be a great explanation for the reader.
- Lines 74, 75 and 76: Reference at least one study.
- Lines from 159 to 171 seem to be some conclusions! If this is the case, please indicate that, e.g. by starting the paragraph with. In conclusion to this part...
In the case this is not the author's conclusion, please reference it.
4.Hypothesis 2 seems to consist also of this variable: 'green technology innovation'. It is not treated as variable. It is clear that this does not affect the results, however, a short explanation could help clarify the reason for its existence within hypothesis 2.
Well, interesting research work!
We would like to thank the reviewers for their positive feedback.
Comment 1: Abstract: improve the consistency between abstract-introduction-problem definition- results-conclusions
Suggestion: include your main hypothesis or research question starting from abstract, research methods and research limitations. A short description including the relationship between variables would be a great explanation for the reader.
Many thanks for the reviewer’s suggestion. We have amended the abstract to read as follows:
Choosing appropriate environmental protection strategies is important to improving enterprises’ economic and environmental performance. Based on the data of China's A-share listed enterprises from 2009–2019 in China, this paper uses the difference-in-differences model to identify the effects of environmental credit constraints on enterprise choice of environmental protection behavior. We find that environmental credit constraints motivate some enterprises to choose active environmental behavior due to the incentive effect of environmental credit constraints on R&D investments. However, some enterprises may adopt evasive strategies because environmental credit constraints increase production costs and debt. State-owned enterprises prefer active environmental protection strategies to address environmental credit constraints, while private enterprises mainly adopt evasive strategies. Environmental credit constraints make high-interest and high-profitability enterprises choose active environmental strategies. Environmental credit constraints generated by enterprises’ evasive environmental behavior in-crease the probability of litigation arbitration cases, and environmental credit system construction in the short term may exacerbate unemployment, which the government needs to pay attention to when developing and implementing a blacklist system for environmental fraud. Although there are limitations in this paper in terms of research objectives and samples, but the results are important for improving the environmental management system and the operating performance of enterprises.
Comment 2: Lines 74, 75 and 76: Reference at least one study.
Many thanks for the reviewer’s suggestion. Based on your suggestion we have selected a paper as a reference study.
- Marquis, C.; Zhang, J.; Zhou, Y. Regulatory uncertainty and corporate responses to environmental protection in China. Cali-fornia Management Review. 2011, 54(1), 39-63.
Comment 3: Lines from 159 to 171 seem to be some conclusions! If this is the case, please indicate that, e.g. by starting the paragraph with. In conclusion to this part...
In the case this is not the author's conclusion, please reference it.
Many thanks for the reviewer’s suggestion. Lines 159 to 171 are a response to the conclusion above, to which we have added at the beginning of the paragraph: In summary.
Comment 4:Hypothesis 2 seems to consist also of this variable: 'green technology innovation'. It is not treated as variable. It is clear that this does not affect the results, however, a short explanation could help clarify the reason for its existence within hypothesis 2.
Many thanks for the reviewer’s suggestion. Hypothesis 2 focuses on how environmental credit constraints affect green technological innovation, expressed as an innovative behavior. And the variable selection indicates the level of green technology innovation, which measures the change of innovation capacity.
Reviewer 3 Report
Comments and Suggestions for Authors
Thank you for the possibility to read and review this very good article. Only some remarks for improving the quality:
1. The abstract needs information on details of the research (sample, country, period, etc.)
2. I didn't find the exact number of investigated companies (in 321 row).
3. Please, check the 402 row. What is the previous article?
4. The article should be ended with the limitations of the research.
5. I didn't find what is the unit of measurement for variables. (in 264-280 rows).
6. Summarizing in 159-166 rows has the effect of repeating, therefore, it is not necessary.
7. The article lacks a theoretical background. We need some theories which can explain the research object, and what theoretical implications of the research results can be named.
8. The article has a strong Chinese context - everything is based on China's market - from the beginning of the text to most of the references. But science is worldwide, so we need broader context and non-Chinese references - authors who have analyzed this research object in the world.
Author Response
Reviewer 3
Thank you for the possibility to read and review this very good article. Only some remarks for improving the quality:
- The abstract needs information on details of the research (sample, country, period, etc.)
- I didn't find the exact number of investigated companies (in 321 row).
- Please, check the 402 row. What is the previous article?
- The article should be ended with the limitations of the research.
- I didn't find what is the unit of measurement for variables. (in 264-280 rows).
- Summarizing in 159-166 rows has the effect of repeating, therefore, it is not necessary.
- The article lacks a theoretical background. We need some theories which can explain the research object, and what theoretical implications of the research results can be named.
- The article has a strong Chinese context - everything is based on China's market - from the beginning of the text to most of the references. But science is worldwide, so we need broader context and non-Chinese references - authors who have analyzed this research object in the world.
We would like to thank the reviewers for their positive feedback.
Comment 1: The abstract needs information on details of the research (sample, country, period, etc.)
Many thanks for the reviewer’s suggestion. We have amended the abstract to read as follows:
Choosing appropriate environmental protection strategies is important to improving enterprises’ economic and environmental performance. Based on the data of China's A-share listed enterprises from 2009–2019 in China, this paper uses the difference-in-differences model to identify the effects of environmental credit constraints on enterprise choice of environmental protection behavior. We find that environmental credit constraints motivate some enterprises to choose active environmental behavior due to the incentive effect of environmental credit constraints on R&D investments. However, some enterprises may adopt evasive strategies because environmental credit constraints increase production costs and debt. State-owned enterprises prefer active environmental protection strategies to address environmental credit constraints, while private enterprises mainly adopt evasive strategies. Environmental credit constraints make high-interest and high-profitability enterprises choose active environmental strategies. Environmental credit constraints generated by enterprises’ evasive environmental behavior in-crease the probability of litigation arbitration cases, and environmental credit system construction in the short term may exacerbate unemployment, which the government needs to pay attention to when developing and implementing a blacklist system for environmental fraud. Although there are limitations in this paper in terms of research objectives and samples, but the results are important for improving the environmental management system and the operating performance of enterprises.
Comment 2: I didn't find the exact number of investigated companies (in 321 row).
Referring to the variable selection criteria of and, enterprise- and city-level indicators (Include 2,659 enterprises, 226 cities) are selected as the control variables.
Comment 3: Please, check the 402 row. What is the previous article?
We are sorry for leading this misunderstanding. Precisely, " the previous article" means "the benchmark regression". This is our misstatement and we have corrected it to" the benchmark regression".
Comment 4: The article should be ended with the limitations of the research.
Thank you for your comment. Your suggestion makes us aware of the deficiencies in the content of the paper. So we added the limitations of the research in the conclusion part.Please see line 836-844 for details (also shown in the image below).
The limitations of this paper are as follows. First, due to the limitations of the sample, the relevant data of small and medium-sized enterprises are difficult to be obtained, and we have to use the data of listed companies, so I have not considered the influence of small and medium-sized enterprises on the research question, which may make the sample have a certain bias. Second, for the limitation of the research object, this paper only aims to study the active environmental protection behavior and evasive environmental pro-tection behavior and does not consider the situation of passive environmental protection behavior, which may lead to the conclusion with certain limitations.
Comment 5: I didn't find what is the unit of measurement for variables. (in 264-280 rows).
Thank you for your comment. The unit of measurement for variables in 279-283 rows have been added.
Comment 6: Summarizing in 159-166 rows has the effect of repeating, therefore, it is not necessary.
Many thanks for the reviewer’s suggestion. In response to your suggestions, we have reconsidered the meaning expressed in 159-166 and revised it. Please see line 162-167 for details (also shown in the image below).
Comment 7: The article lacks a theoretical background. We need some theories which can explain the research object, and what theoretical implications of the research results can be named.
Thank you for your comment. Your suggestion makes us aware of the deficiencies in the content of the paper. So we added some theories in the theoretical hypothesis part.Please see line 174-179 for details (also shown in the image below).
Comment 8: The article has a strong Chinese context - everything is based on China's market - from the beginning of the text to most of the references. But science is worldwide, so we need broader context and non-Chinese references - authors who have analyzed this research object in the world.
Many thanks for the reviewer’s suggestion. First, we study the Chinese issue, and China, as one of the largest developing countries, China's experience can be learned from. Secondly, in terms of literature, we have also added relevant English literature to supplement the article.
Reviewer 4 Report
Comments and Suggestions for Authors
Thank you for receiving the paper titled: Environmental Credit Constraints and the Enterprise Choice of Environmental Protection Behavior to review.
The paper consists of 24 pages and 50 references, which from a formal point of view allows it to be considered a scientific article. After carefully reading this interesting paper, I have a few comments, suggestions and advice.
In the introduction, the authors wrote about the problem of population growth, economic development and the resulting actions taken. I would suggest entering a quote. In a later sentence, the authors refer to the situation in China and there is a footnote [1], therefore the part related to the general problem of population growth, increased pollution, etc. should also include a citation. I suggest you read two articles that present this issue on a global scale.
In the literature review, the authors write that many researchers have dealt with issues (legal, social, political environment), so I propose to include one more quote as an example. I suggest to read the paper: Adaptation strategy on regulated markets of power companies in Poland. Energy & Environment, 30(1), 3-26.
I would like that Authors will explain what is active and what is evasive behaviour. Usually, the opposite of active strategies or behaviors is passive. The authors use the term evasive. I suggest to read papers concerning with active and passive adoptation. New technologies and innovative solutions in the development strategies of energy enterprises. HighTech and innovation Journal, 1(2), 39-58.
What does "the parallel trend hypothesis" mean? Please show the verification of 3 hypotheses.
Please write conclusions and recommendations instead of conclusions and suggestions. In the world of science, it is common to use recommendations.
Author Response
Reviewer 4
Thank you for receiving the paper titled: Environmental Credit Constraints and the Enterprise Choice of Environmental Protection Behavior to review.
The paper consists of 24 pages and 50 references, which from a formal point of view allows it to be considered a scientific article. After carefully reading this interesting paper, I have a few comments, suggestions and advice.
In the introduction, the authors wrote about the problem of population growth, economic development and the resulting actions taken. I would suggest entering a quote. In a later sentence, the authors refer to the situation in China and there is a footnote [1], therefore the part related to the general problem of population growth, increased pollution, etc. should also include a citation. I suggest you read two articles that present this issue on a global scale.
In the literature review, the authors write that many researchers have dealt with issues (legal, social, political environment), so I propose to include one more quote as an example. I suggest to read the paper: Adaptation strategy on regulated markets of power companies in Poland. Energy & Environment, 30(1), 3-26.
I would like that Authors will explain what is active and what is evasive behaviour. Usually, the opposite of active strategies or behaviors is passive. The authors use the term evasive. I suggest to read papers concerning with active and passive adoptation. New technologies and innovative solutions in the development strategies of energy enterprises. HighTech and innovation Journal, 1(2), 39-58.
What does "the parallel trend hypothesis" mean? Please show the verification of 3 hypotheses.
Please write conclusions and recommendations instead of conclusions and suggestions. In the world of science, it is common to use recommendations.
We would like to thank the reviewers for their positive feedback.
Comment 1: In the introduction, the authors wrote about the problem of population growth, economic development and the resulting actions taken. I would suggest entering a quote. In a later sentence, the authors refer to the situation in China and there is a footnote [1], therefore the part related to the general problem of population growth, increased pollution, etc. should also include a citation. I suggest you read two articles that present this issue on a global scale.
Thank you for your comment.As suggested by the reviewer, we have added more references to support this idea.
- Kaiser, J.; Lerch, M. Sedimentary faecal lipids as indicators of Baltic Sea sewage pollution and population growth since 1860 AD. Environmental Research. 2022, 204, 112305.
[2] Rehman, A.; Ma, H.; Ozturk, I. Sustainable development and pollution: The effects of CO2 emission on population growth, food production, economic development, and energy consumption in Pakistan. Environmental Science and Pollution Research. 2022, 1-12.
Comment 2: In the literature review, the authors write that many researchers have dealt with issues (legal, social, political environment), so I propose to include one more quote as an example. I suggest to read the paper: Adaptation strategy on regulated markets of power companies in Poland. Energy & Environment, 30(1), 3-26.
Thank you for your comment. As suggested by the reviewer, we have added “Adaptation strategy on regulated markets of power companies in Poland” this article to support this idea.
Comment 3: I would like that Authors will explain what is active and what is evasive behaviour. Usually, the opposite of active strategies or behaviors is passive. The authors use the term evasive. I suggest to read papers concerning with active and passive adoptation. New technologies and innovative solutions in the development strategies of energy enterprises. HighTech and innovation Journal, 1(2), 39-58.
That's a very good question. Usually, the opposite of active strategies or behaviors is passive, but we think passive strategies or behaviors should be distinct from evasive behaviors. First, passive environmental protection behavior. Enterprises adopting this behavior will have to pay pollution control fees to improve their production modes because of environmental credit constraints, which is bound to further increase production costs. Such enterprises may choose to reduce their production scale but continue to use old manufacturing production processes, reducing pollutant emissions by reducing economic output. Second, evasive environmental protection behavior. Enterprises adopting this behavior have a negative attitude toward environmental credit constraints and will take advantage of the differences in the environmental regulation intensity in different regions to avoid environmental penalties or reduce environmental costs through production transfers.
Evasive environmental protection behavior may be a more extreme behavior in response to environmental regulation, which can be contrasted with active environmental protection behaviors, whereas passive behaviors and active behaviors have a strong correlation, and often one behavior can be used to reflect the other.
We add “New technologies and innovative solutions in the development strategies of energy enterprises”this article to support our idea.Please see line 189-190 for details (also shown in the image below).
Comment 4: What does "the parallel trend hypothesis" mean? Please show the verification of 3 hypotheses.
We are sorry for leading this misunderstanding. Precisely, " the parallel trend hypothesis" means "requirements for parallel trends". This is our misstatement and we have corrected it to" the result".
Comment 5: Please write conclusions and recommendations instead of conclusions and suggestions. In the world of science, it is common to use recommendations.
Many thanks for the reviewer’s suggestion.Based on the review comments, we have changed the “suggestions” to “recommendations”.
Reviewer 5 Report
Comments and Suggestions for Authors
Dear Authors,
thank you for interesting and valuable research. However, I do have a few suggestions to improve your paper.
1. What do you mean by a word 'departures' in line 67?
2. Please support the following statements by references: "Although existing studies" (line 74), many scholars have conducted (line 94), some scholars have studied (line 139), many scholars have studied (line 159) and many more. Please support your sayings by scientific literature.
3. Please reconsider yours first hypothesis. I find it hardly a scientific one. Could you elaborate and justify this hypothesis to a greater extent?
4. Why did you choose such control variables? I strongly believe that it is not enough to give 39 and 40 references to explain your choice.
5. What are the limitations of your research? It would be good to have this spart in your conclusions part.
6. What future research are needed to understand enterprises behavior better in regard of environmental credit constrains?
Comments on the Quality of English LanguageMinor editing of English language required.
Author Response
Reviewer 5
Dear Authors,
thank you for interesting and valuable research. However, I do have a few suggestions to improve your paper.
- What do you mean by a word 'departures' in line 67?
- Please support the following statements by references: "Although existing studies" (line 74), many scholars have conducted (line 94), some scholars have studied (line 139), many scholars have studied (line 159) and many more. Please support your sayings by scientific literature.
- Please reconsider yours first hypothesis. I find it hardly a scientific one. Could you elaborate and justify this hypothesis to a greater extent?
- Why did you choose such control variables? I strongly believe that it is not enough to give 39 and 40 references to explain your choice.
- What are the limitations of your research? It would be good to have this spart in your conclusions part.
- What future research are needed to understand enterprises behavior better in regard of environmental credit constrains?
7.Minor editing of English language required.
We would like to thank the reviewers for their positive feedback.
Comment 1: What do you mean by a word 'departures' in line 67?
We are sorry for leading this misunderstanding. Precisely, " departures" means "separations". This is our misstatement and we have corrected it to" the introduction".
Comment 2: Please support the following statements by references: "Although existing studies" (line 74), many scholars have conducted (line 94), some scholars have studied (line 139), many scholars have studied (line 159) and many more. Please support your sayings by scientific literature.
Many thanks for the reviewer’s suggestion. As suggested by the reviewer, first, we have added more references to support "Although existing studies" (line 74), many scholars have conducted (line 94), some scholars have studied (line 139).Secondly, because the content of line 159 has been changed, the relevant literature is not cited. Please see line 162-167 for details (also shown in the image below).
[11] Marquis, C.; Zhang, J.; Zhou, Y. Regulatory uncertainty and corporate responses to environmental protection in China. Cali-fornia Management Review. 2011, 54(1), 39-63.
[12] Borowski, P. F. Adaptation strategy on regulated markets of power companies in Poland. Energy & Environment. 2019, 30(1), 3-26.
[31] Acheampong, A. O. Governance, credit access and clean cooking technologies in Sub-Saharan Africa: Implications for energy transition. Journal of Policy Modeling. 2023, 45(2), 445-468.
Comment 3: Please reconsider yours first hypothesis. I find it hardly a scientific one. Could you elaborate and justify this hypothesis to a greater extent?
Many thanks for the reviewer’s suggestion. We have modified hypothesis 1 in accordance with your suggestions, revising hypothesis 1 as follows: Environmental credit constraints can produce active environmental protection behavior for enterprises while reinforcing evasive environmental protection behavior.
Comment 4: Why did you choose such control variables? I strongly believe that it is not enough to give 39 and 40 references to explain your choice.
Many thanks for the reviewer’s suggestion. We explained our choice of control variables by specifying them according to the reviewers' suggestions, thus explaining our reasons for choosing them. Please see line 310-343 for details.
Comment 5: What are the limitations of your research? It would be good to have this spart in your conclusions part.
Thank you for your comment. Your suggestion makes us aware of the deficiencies in the content of the paper. So we added the limitations of the research in the conclusion part.Please see line 836-844 for details (also shown in the image below).
The limitations of this paper are as follows. First, due to the limitations of the sample, the relevant data of small and medium-sized enterprises are difficult to be obtained, and we have to use the data of listed companies, so I have not considered the influence of small and medium-sized enterprises on the research question, which may make the sample have a certain bias. Second, for the limitation of the research object, this paper only aims to study the active environmental protection behavior and evasive environmental pro-tection behavior and does not consider the situation of passive environmental protection behavior, which may lead to the conclusion with certain limitations.
Comment 6: What future research are needed to understand enterprises behavior better in regard of environmental credit constrains?
Thank you for your comment. Your suggestion makes us aware of the deficiencies in the content of the paper. So we added future research of the research in the conclusion part. Please see line 845-850 for details.
Future research can be further carried out in the following aspects: First, expand the research scope to longer periods and then observe whether the findings of this paper are still valid. Second, in subsequent studies, data from small and medium-sized enterprises can be added to the study to increase the sample size and sample range, so as to reduce sample bias. Third, further research on the impact of environmental credit constraints on enterprises' environmental investments and other aspects.
Comment 7: Minor editing of English language required.
Thanks for your suggestion. We have tried our best to polish the language in the revised manuscript.
Round 2
Reviewer 3 Report
Comments and Suggestions for Authors
Thank you for improving the article. We see perfect results.
Reviewer 4 Report
Comments and Suggestions for Authors
Dear Authors,
Thank you for explaining the authors' point of view related to the evasive strategy. The authors explained their understanding of active, passive and evasive strategies.
Thank you for taking into account my comments and suggestions. Authors made every effort to raise the standard of their paper.
The authors have increased the literature review so that potential readers can benefit from a wide range of related publications.
I have no further questions or comments, the paper can be published in this form.